# Thermotropic Liquid-Crystalline Materials Based on Supramolecular Coordination Complexes

**Bruno Therrien** 

Institut de Chimie, Université de Neuchâtel, Avenue de Bellevaux 51, CH-2000 Neuchâtel, Switzerland; bruno.therrien@unine.ch

**Abstract:** Liquid crystals are among us, in living organisms and in electronic devices, and they have contributed to the development of our modern society. Traditionally developed by organic chemists, the field of liquid-crystalline materials is now involving chemists and physicists of all domains (computational, physical, inorganic, supramolecular, electro-chemistry, polymers, materials, etc.,). Such diversity in researchers confirms that the field remains highly active and that new applications can be foreseen in the future. In this review, liquid-crystalline materials developed around coordination complexes are presented, focusing on those showing thermotropic behavior, a relatively unexplored family of compounds.

**Keywords:** liquid crystals; thermotrope; supramolecular chemistry; coordination complexes; lyotropes

---

## 1. Introduction

Liquid-crystals are known for over a century, and despite being part of our everyday life, basic research on liquid-crystalline materials remains highly attractive [1]. Indeed, new applications are emerging, in which the particular properties of liquid-crystals are exploited [2,3]. Accordingly, liquid crystals incorporating metal-based entities are showing great promises [4–10]. The introduction of additional intermolecular interactions, such as metal–metal or metal–ligand, into the supramolecular structure of the liquid-crystalline material, can modulate the properties, and ultimately, provide new opportunities.

Liquid-crystalline materials are classified into thermotropes and lyotropes [11]. In thermotropic liquid crystals, the mesophases are induced by the temperature, while in lyotropic liquid-crystals the mesophases are controlled by the concentrations of the different components in a single or mixture of solvents. Liquid-crystalline materials found in nature, such as lipids and membranes, are lyotropes. On the other hand, those involved in electronic devices are more likely to be thermotropes. In the case of metal-based liquid-crystalline materials, both types can be found in the literature [4–10], and some can even be amphotropes [11].

Among metal-based liquid-crystalline materials, those exploiting supramolecular coordination complexes have not yet found a commercial application, however, they are very interesting and they offer great potentials in biology [12–16] and material sciences [17–19]. Moreover, and despite having well-established synthetic strategies, the number of papers dealing with supramolecular coordination complexes as liquid-crystalline promoters remains low. This might be associated to the complexity of such systems, in which additional interactions are involved, thus increasing the difficulty of predicting and interpreting the corresponding mesophases. Nevertheless, owing to the development of our understanding on the molecular organization within mesophases, based on experiences as well as on new computational models, we should see more of these hybrid supramolecular liquid-crystalline materials in the future.

## 2. Lyotropic Liquid-Crystalline Self-Assembled Coordination Complexes

Incorporation of supramolecular coordination complexes in liquid-crystalline materials was first explored by Praefcke and Usol'tseva in the early 1990s. The lyomesomorphic behavior of large metalla-assemblies were showing nematic phases in alkanes [20]. In the case of the chloro-and bromo-bridged metalla-cycles (Figure 1), the nematic phase was stabilized upon addition of 2,4,7-trinitrofluorenone (TNF), suggesting intercalation of TNF between the disk-shaped columnar stacks. Following this initial study, analogous systems built from either palladium or platinum metal centers were prepared and their liquid-crystalline properties examined [21–25], showing similar organization in the lyotropic mesophases.

**Figure 1.** Early supramolecular coordination complexes showing liquid-crystalline properties [20].

Later on, analogous platinum Schiff base pyridyl type metalla-cycles were synthesized by MacLachlan and his coworkers [26]. In the series, the 2-hexyldecyl derivative (Figure 2) showed the most interesting liquid-crystalline properties. In non-polar organic solvents, lyotropic mesophases were observed for the tetranuclear metalla-cycle. As demonstrated by the authors, a supramolecular aggregation of individual Pt-based metalla-cycle into columnar arrays was responsible for the liquid-crystalline properties.

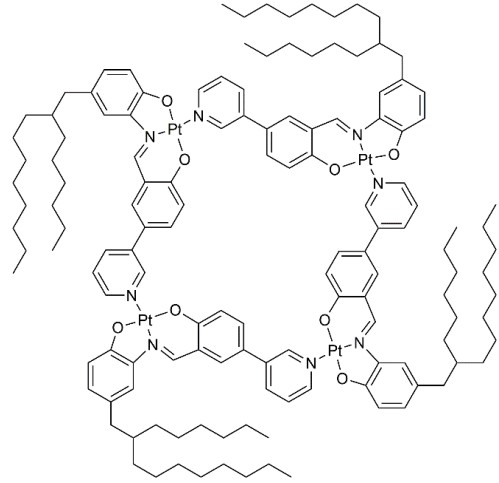

**Figure 2.** A tetranuclear metalla-cycle with lyotropic liquid-crystalline behavior [26].

In a similar manner, Gin and co-workers have prepared dinuclear-Pt mesogenic metalla-cycles incorporating bis-pyridyl linkers [27]. The hexafluorophosphate salt (Figure 3) possesses a thermotropic columnar hexagonal (Col$_H$) liquid-crystalline phase, which can be swollen by polar solvents, to generate a lyotropic liquid-crystalline phase. Upon irradiation, conversion of the *azo* group to the *cis* isomer was observed, thus provoking a disorder in the liquid-crystalline phase. Photo-polymerization of neighboring metalla-cycles increases the stability of the mesophases, with however, a similar rate of conversion of the *azo* groups.

**Figure 3.** A dinuclear platinum-based metalla-cycle with Col$_H$ liquid-crystalline properties in polar solvents [27].

Trinuclear gold metalla-cycles have been used to generate columnar liquid-crystalline materials [28]. The presence of six alkyl chains at the periphery of the pyrazolato-gold complex (Figure 4) controls the formation of columnar stacks in the solid and liquid-crystalline states; Both showing an hexagonal symmetry. Interestingly, the functional groups (R, R') can be equivalent or not, thus allowing the formation of two isomers. The proportion and nature of the two isomers influence the transition temperatures, and accordingly, the arrangement of the trinuclear assemblies in the lyotropic mesophases.

**Figure 4.** Trinuclear pyrazolato-gold metalla-cycles with columnar arrangement [28].

Addition of water to discrete septuple columnar stacks can induce lyotropic liquid-crystalline mesophases [29]. The columnar stacks were synthesized in solution by the self-assembly of a bis-pyridyl linker, tris(4-pyridyl)-2,4,6-triazine (tpt), triphenylene, and the palladium complex (en)Pd(NO$_3$)$_2$ (en = ethylenediamine), see Scheme 1. Assemblies composed of six bis-pyridyl linkers, four tpt panels, twelve (en)Pd corners, and three triphenylene intercalated guests, were isolated. The presence of

water-soluble side chains on the bis-pyridyl linkers was crucial for the generation of the mesophases and for keeping fluidity to the supramolecular system.

**Scheme 1.** Building-blocks used to synthesize septuple columnar stacks in solution [29].

## 3. Thermotropic Liquid-Crystalline Self-Assembled Coordination Complexes

Metal-based clusters have been used as central cores to generate mesomorphic materials. Functionalization of the peripheral ligands has allowed a bottom-up approach to prepare liquid-crystalline materials. The choice of the cluster dictates the type of the functionalized ligands to be used. For the manganese cluster, $[Mn_{12}O_{12}(RCO_2)_{16}(H_2O)_4]$, functionalized carboxylic acid derivatives are needed (Figure 5A). The $Mn_{12}O_{12}$ clusters show mesophases with cubic or smectic phases. The magnetic properties of the cluster are retained in the mesophases, thus providing magnetic liquid-crystalline materials [30]. Similarly, to insert mesogenic arms to sawhorse-type dinuclear ruthenium complexes, carboxylic acid derivatives are needed (Figure 5B). In these systems, the cyanobiphenyl-based poly(arylester) dendron was used to induce mesomorphic properties [31]. Smectic A and nematic phases were observed, according to the generation of the dendrimers.

**Figure 5.** Functionalized carboxylic acid derivatives grafted to manganese (**A**) and ruthenium (**B**) metallic-cores [30,31].

In the case of the cupper cluster $[Cu_4I_4]$, diphenyl-phosphine ligands (Figure 6A) were linked to the cubic tetrametallic core, thus introducing four mesogenic side-chains [32]. The supramolecular system shows a smectic A phase from room temperature to 100 °C. In the case of the spherical $Pd_{12}L_{24}$ framework (L = bis-pyridyl ligand), functionalized bis-pyridyl connectors (Figure 6B) were used to

ensure stability of the $Pd_{12}$ core, and to insert mesogenic arms [33]. Interestingly, the free ligands show thermotropic behavior, while the cluster-based systems have lyotropic properties.

**Figure 6.** Functionalized diphenyl-phosphine (**A**) and bis-pyridyl (**B**) ligands coordinated to clusters [32,33].

Rectangular columnar arrangement has been observed for nickel, palladium, and copper tetranuclear metalla-cycles (Figure 7). In these systems, a large inner cavity ($\approx 9$ Å in diameter) is observed in the metalla-cycle [34]. The isotropic temperature of the metalla-cycles was higher than the metal free cycle, and well below the temperature of decomposition. The nature of the peripheral chains and the choice of the metal influence the thermotropic behavior [35]. Moreover, the cavity can be exploited to accommodate guest molecules, and accordingly, to offer another alternative for fine tuning the liquid-crystalline properties.

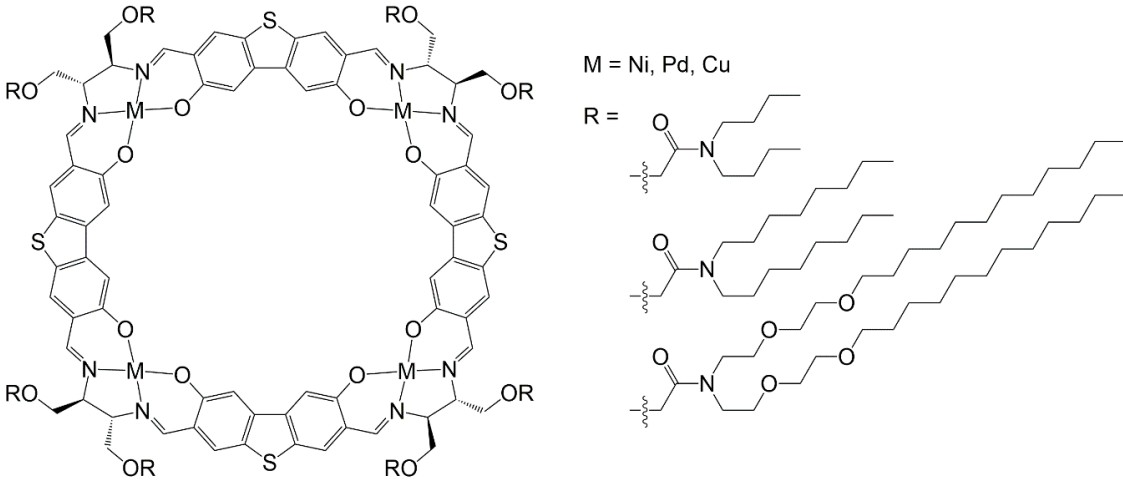

**Figure 7.** Rectangular columnar systems with thermotropic liquid-crystalline properties [34].

Bis-pyridyl linkers can also be used to assemble arene-ruthenium metalla-rectangles [36]. Addition of 1,4-di(4-pyridinyl)-benzene poly(arylester) derivative to the dinuclear arene-ruthenium complex $[Ru_2(p\text{-cymene})_2(donq)][DDS]_2$ (donq = dihydroxynaphthoquinone, DDS = dodecyl sulfate) generates a tetranuclear metalla-cycle, isolated as its DDS salt (Scheme 2). The presence of four dendritic arms bearing cyano-biphenyl end-groups ensures mesomorphic properties above 50 °C. Both compounds, the bis-pyridyl linker and the metalla-cycle, possess smectic phases with a multilayered organization [37].

**Scheme 2.** Synthesis of an arene-ruthenium metalla-cycle from a functionalized dendritic bis-pyridyl linker [37].

The host-guest chemistry of arene-ruthenium metalla-assemblies is well-established [38,39], and arene-ruthenium metalla-prisms and metalla-rectangles have been used to encapsulate pyrenyl-functionalized guests [40]. In the case of the pyrenyl-functionalized dendromesogenic guest encapsulated in a tetranuclear arene-ruthenium metalla-cycle (Figure 8), thermotropic liquid-crystalline properties were observed [41]. The guest alone is showing a smectic A phase, while the host-guest system possesses a cubic phase. The multi-component arrangement is highly segregated, suggesting a multi-layered structure involving metalla-cycles, dodecyl sulfates, and the side-arms of the pyrenyl-functionalized dendrons.

**Figure 8.** Pyrenyl-functionalized dendrimer encapsulated in an arene ruthenium metalla-cycle [41].

## 4. Suppression of Liquid-Crystalline Properties by Self-Assembled Coordination Complexes

Encapsulation of guest molecules in metalla-assemblies offers great potentials in molecular recognition [42], transport [43], and protection of guest molecules [44], as well as in cavity-controlled reaction (molecular flask) [45]. Moreover, the presence of guest molecules in the cavity of a supramolecular coordination complex can modify its geometry and properties. Accordingly, liquid-crystalline behavior can be modulated upon host–guest interactions. Indeed, encapsulation of pyrenyl-functionalized dendrimers in the cavity of a hexanuclear arene ruthenium metalla-prism has showed the suppression of the liquid-crystalline properties of the organic compound (guest) [46]. The pyrenyl-functionalized poly (arylester) dendron (Figure 9A) shows an unidentified liquid-crystalline behavior, and no cytotoxicity to cancer cells (A2780 and A2780cisR). On the other hand, the host–guest system is cytotoxic, with $IC_{50} < 3 \, \mu M$ on these two cancer cell lines, with however, no liquid-crystalline behavior.

**Figure 9.** Two pyrenyl-functionalized liquid-crystalline compounds (**A**,**B**) showing suppressed liquid-crystalline properties after encapsulation in a coordination complex [46,47].

Similarly, a porphyrin-based tetragonal prism has been used to encapsulate a pyrenyl-functionalized guest compound [47]. The pyrenyl derivative (Figure 9B) possesses between 23 and 107 °C a smectic A phase. Upon addition of the platinum-based tetragonal prismatic host, the mesomorphic properties of the guest are lost. However, adding coronene or pyrene as competing guests, re-established the liquid-crystalline properties of the pyrenyl-functionalized compound, thus offering a switch-on switch-off control over the liquid-crystalline properties.

## 5. Conclusions

The field of liquid-crystalline materials is shifting toward more complexed systems, in which multiple components are used to generate highly organized supramolecular arrangements. These sophisticated materials are expected to have an impact in various fields, such as in engineering (molecular electronics, photonics, high mechanical strength fibers, light modulators, lasers), energy (battery electrolytes), healthcare (artificial membranes, drug delivery, gene therapy), environment (biocompatible plastics), chemistry (surfactants, detergents, elastomers, gels), separation technology (sensors), informatics (intelligent switches), catalysis, and others [1–4,48–59].

A pillar of supramolecular chemistry is coordination chemistry [60], and therefore, it is not surprising to see more and more supramolecular coordination complexes being incorporated within liquid-crystalline materials. However, as illustrated in this short review, examples dealing with

thermotropic liquid-crystalline coordination complexes remain scarce. Nevertheless, when considering the number of active groups in the field of coordination-driven self-assemblies [61–64], and the exciting applications that can be foreseen for the next generation of liquid-crystalline materials, this trend should be reversed in a few years.

**Funding:** B.T. thanks the Swiss National Science Foundation (grant No 200021-162361) for financial support.

**Conflicts of Interest:** The author declares no conflict of interest.

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
