# Peer review of "Thermotropic Liquid-Crystalline Materials Based on Supramolecular Coordination Complexes"

_inorganics, doi:10.3390/inorganics8010002_

Round 1

Reviewer 1 Report

The short review by Therrien focus on the thermotropic liquid-crystalline materials based on supramolecular coordination complexes. However, I think listing the examples of the materials and their properties is not enough for a review. I couldn’t recommend the manuscript for publishing in Inorganics unless additional two sections can be added at the end of this review.

1. The potential applications of thermotropic liquid-crystalline materials based on supramolecular coordination complexes.

2. The future outlook of developing such materials should also be discussed.

Author Response

We would like to thank the reviewer for his (her) suggestion, but I don't think that adding two sections at the end of the manuscript is appropriate here, especially when considering the various applications and the difficulty to mention everything. Moreover, several excellent reviews are giving such perspectives and outlooks, therefore, a new sentence has been added in the conclusion to list potential applications, and accordingly, 12 new references (reviews) have been Added in the revised version.

The new sentence is: These sophisticated materials are expected to have an impact in various fields, such as in engineering (molecular electronics, photonics, high mechanical strength fibers, light modulators, lasers), energy (battery electrolytes), healthcare (artificial membranes, drug delivery, gene therapy), environment (biocompatible plastics), chemistry (surfactants, detergents, elastomers, gels), separation technology (sensors), informatics (intelligent switches), catalysis and others [1-4, 48-59].

Reviewer 2 Report

In this manuscript, the author describes the use of coordination complexes to more complex supramolecular coordination assemblies in the field of liquid-crystalline materials field. Research in the domain is little explored. To gather these examples presents a real interest, the complementary supramolecular interactions lead to a modulation of  the properties and opens potential ways to new applications. Overall, the manuscript is very well written and the examples are judiciously chosen. Therefore, I would support the acceptance of the article in Inorganics.

Anecdotal changes have to be made:

- the negative charge of the anion figure 3 is missing

-There is a no void space  line 121 between aryl and ester

Author Response

We thank the reviewer, and the two corrections have been done (figure 3 and typo).

Round 2

Reviewer 1 Report

I still think such information is necessary for a review article. I am sorry I cannot give a positive answer if the author refuse to revise the manuscript in a suitable way.